# Succession Planning Leadership Model for Nurse Managers in Hospitals: A Narrative Review

**DOI:** 10.3390/healthcare11040454

**Published:** 2023-02-04

**Authors:** Kurniawan Yudianto, Nanan Sekawarna, F. Sri Susilaningsih, Vimala Ramoo, Irman Somantri

**Affiliations:** 1Faculty of Nursing, Padjadjaran University, Sumedang 45363, Indonesia; 2Faculty of Medicine, Universitas Islam Bandung, Bandung 40116, Indonesia; 3Faculty of Medicine, University of Malaya, Kuala Lumpur 50603, Malaysia

**Keywords:** leadership, nursing, succession planning

## Abstract

The high number of nursing staff in Indonesia requires optimal management skills, one of which comes from the leadership domain. The succession planning program can be an option to prepare nurses who have leadership potential to carry out a management function. This study aims to identify the nurse succession planning model and its application in clinical practice. This study uses a narrative review of the literature approach. Article searches were carried out using electronic databases (PubMed and Science Direct). Researchers obtained 18 articles. Three main themes emerged: (1) the factors that influence the efficient implementation of succession planning, (2) the benefits of succession planning, and (3) the implementation of succession planning in clinical practice. Training and mentoring related to leadership, support from human resources, and adequate funding are the main factors in implementing effective succession planning. Succession planning also can help nurses find competent leaders. However, in its application in clinical practice, the process of recruitment and planning for nurse managers that occurs in the field is not optimal so that succession planning must exist and be integrated with organizational needs and provide guidance and assistance for the younger generation who will become leaders in the future.

## 1. Introduction

Leaders have an important role in driving changes in nursing, in both the culture and in clinical practice. Nursing leaders have a key role in creating a positive practice environment in support of patient safety programs, especially at the level of nurse managers [1]. The nurse manager is in a key position to influence success in the ward. Thus, the competence of the nurse manager is expected to be able to maintain the quality of nursing care to create job satisfaction for the rest of the nurse staff [1].

Based on data from the World Health Organization, almost 50% of health workers are nurses [2]. While based on data from Indonesian Health Ministry, the number of nurses in Indonesia in 2021 is the largest compared to other health workers, reaching 511,191 staff [3]. The role of a nurse is also needed to achieve Sustainable Development Goal 3 on health and well-being [2].Given the large number of nursing staff and how essential nursing role is, good leadership is paramount in enhancing nurses’ performance [4].

In enhancing good leadership, nurses must possess good nursing care plans and patient outcomes, and nurse managers play an important role in enhancing the competency of their staff. Nurse managers should focus on improving managing and organizing the work of nurses by possessing managerial competence [5]. Many nurse managers will retire, and that makes leadership in nursing meet their criteria [6]. Nurse managers who are about to retire have provided knowledge about the competencies and skills needed by future nurse leaders, but current leadership development is still limited [6]. A good recruitment and selection process is one of the factors that influence managerial competence of nurse managers in Indonesia [7]. Succession planning can be an option to achieve good recruitment and selection process. Even if the healthcare organization does not have the effort to perform succession planning, the nursing department must design and implement its leadership succession plan, because the benefits will outweigh the costs, both in the short- and long-term [8].

Succession planning is one way to identify the leadership capacity possessed by junior staff, as well as a clear career path; this must be in line with the development planned for leaders of the future. Staff who will take up the leadership baton must be ready to accept the leadership relay or be seen as having long-term leadership potential [9]. The review aimed to identify the nurse succession planning model and its application in clinical practice.

## 2. Materials and Methods

This study uses the narrative review method. The researcher chose this method because the purpose of the study was to develop a model of succession planning for nurses in hospitals and to analyze the effect of succession planning on the competence of nurse managers. Narrative review is a review that aimed to identify and summarize what has been previously published, avoid duplication, and look for new fields of study that have not been discussed [10].

Article searches were carried out using electronic databases, namely PubMed (2009–2022) and Science Direct (2009–2022). A literature search used keywords and Boolean operators adapted to the MeSH term: [(Succession Planning OR Succession Planning Model) AND (Nursing OR Nursing Leadership)]. The researcher also added a feature to filter articles that were in English and had full text. In addition, researchers also excluded articles that only discussed succession planning in general or not in the realm of nursing.

### Selection of Sources of Evidence

The author carries out a study selection process following the PRISMA flow diagram [Fig healthcare-11-00454-ch001]: (1) identifying articles based on keywords; (2) identifying duplicates; (3) filtering by title; (4) filtering based on abstracts; (5) filtering after reading the full text. Data is processed or extracted manually from research results using tables and analyzed based on content.

## 3. Results

After screening 1425 articles from PubMed (n = 157) and Science Direct (n = 1268) by reading the title and abstract, researchers obtained 18 articles relevant to this research. Important things found in these articles are shown in Table 1. The articles analyzed had various designs, namely literature review (n = 11), quantitative-exploratory (n = 1), quasi-experiments (n = 1), case studies (n = 2), pilot studies (n = 1), explorative descriptive (n = 1), and analytic descriptive (n = 1).

## 4. Discussion

This study found that succession planning has many benefits if we perform it effectively. We should find out factors that influence the effectiveness of succession planning and how succession planning performs in clinical practice. The findings are discussed based on the following themes.

### 4.1. Factors Influencing the Effectiveness of Succession Planning

Effective succession planning must exist as a continuum within an organization. Several main factors become the focus of researchers to make efficient succession planning. Nurse managers must be given training and mentoring related to leadership in the process of implementing succession planning practices [12,18]. Before carrying out education, training, and mentoring, it is better to identify leadership talents in prospective nurse managers first [26]. This is done so that the succession planning program is more efficient because only prospective nurse managers who have high leadership potential are given training and mentoring.

After talent identification and training and mentoring, succession planning practices must also be supported by existing human resources, namely from management and staff [12]. Nurse managers and nurse staff must be involved in implementation of succession planning. That needs encouragement from nurse managers from each room to each staff. Environmental factors, especially the allocation of resources, are important factors in the development and sustainability of an effective succession planning program [23]. This must be done to build the trust of potential participants to take part in this succession planning program. Prospective participants must have confidence that this program will benefit them.

The existence of a training plan and support from existing human resources is not sufficient to carry out succession planning. The program must also be supported by planned funding [12]. The funds are used as capital to organize training and mentoring for prospective nurse managers who have expertise. In addition, existing funds can also be used to provide incentives for nurse managers so that participants are more interested in participating in this succession planning program [26]. That planned fund also can be used for the promotion of the program.

Succession planning includes strategic planning, analysis of current and future leadership, identification of potential, development of leadership that needs organizational commitment and resource allocation, a proactive and visionary leadership style approach, and an environment of mentoring and coaching [29].

These are the main factors that can support the succession planning process into an effective and efficient program. However, there are several other supporting factors in the success of this succession planning program. This needs to be promoted with careful preparation in the succession planning process so that this program is more socialized and attracts prospective managers who will participate [26]. The succession planning process which involves many parties also requires researchers to pay attention to communication, collaboration, coordination, and transparency between academics [18,26]. This is intended to minimize misunderstandings that lead to the failure of this succession planning program.

### 4.2. Benefits of Succession Planning

The succession planning program is not carried out without reason, but as an effort to increase leads for prospective nurse managers. Nursing staff felt a positive impact related to the succession planning program carried out at their hospitals in Australia and the United States [11,23]. This perceived benefit is not only for the individual nurse but also for future nursing organizations and nursing leadership. Succession planning is a strategic way to identify and develop individuals with high potential for leadership positions who can contribute to the future nursing leadership path [16]. Individuals with high potential for these leadership positions will be prepared to become a new generation of nurse leaders so that the future nurse leadership path will be better [13].

A study in United States showed that nurses who participate in a succession planning program had an increased readiness to be nurse managers. This can be seen in the widespread understanding of the risks of burnout and ongoing stress-related turnover among nurse managers. Healthcare organizations must nurture a new generation of nurses ready to lead teams in the provision of high-quality care [30]. The knowledge of nurse managers regarding succession planning practice, leadership practice, and organizational resilience also increased after the succession planning section [31].

Succession planning can increase recruitment and retention which reinforces the view that individuals are important assets in organizational success [20].This is also supported by a study from the United States which shows that when succession planning program is carried out strategically, it can increase retention rates, increase staff involvement, and improve financial performance [32]. The existence of succession planning will also make health service organizations, especially nursing, survive leadership changes, support organizational goals, and provide opportunities for employees to develop their potential [21]. Succession planning can increase the number of leaders who are ready to take on important positions [24]. Thus, it can be said that succession planning allows nursing institutions to be proactive about sudden leadership vacancies [19].

The succession planning model is the basis for nurse managers to achieve and uphold quality leadership in nursing [19]. Perceptions of prospective nurse managers about leadership and management competencies also increased significantly after participating in succession planning programs [12].Succession planning programs for formal nurse managers have also been shown to increase the competency and retention rates of nurse managers [16]. Aspiring nurse managers will also feel more confident in becoming leaders of an organization [11]. A study from Australia also showed that succession planning can increase nurses’ confidence in carrying out the role of unit manager and in their management skills [6].

Effective implementation of a formal nurse manager succession planning program can also reduce the cost and time of the recruitment process [14]. Leadership development through formal succession planning programs also influences employee satisfaction and can improve the quality of an organization [21]. The quality of an organization is closely related to a healthy workplace environment. Good leadership influences a healthy workplace environment [28].

### 4.3. Application of Succession Planning in Clinical Practice

Succession planning in nursing is closely related to the recruitment process for the nurse manager. In general, the nurse manager is chosen based on the expertise and length of time the nurse has worked in a room [33]. One of the hospitals studied in Indonesia does not have a standard that should implement formal education, and furthermore, the planning function of nurse managers that occurs in the field in the development of formal nurse education is not optimal [22]. Nurse managers also lack the leadership training experience that would be required for the role of nurse manager [33]. Nurses had no knowledge related to the succession planning process due to a lack of education or training preparation before they assumed the position of nurse manager [12]. Most heads of rooms only understand the duties and responsibilities of manager nurses based on the habits and experience of previous seniors [27].

Some nurse managers already understand that succession planning can develop the potential of nurses to become leaders, increase nurse career clarity, help maintain intellectual capital and also help achieve an organizational vision [26]. However, there are other challenges that nurse managers face in succession planning practice, namely: overhaul, the tendency to have conflict, lack of incentives for nurse managers, and a progress rating system for nursing [26]. The lack of a framework in succession planning practice is also a challenge [21]. Given these challenges, a leader must make decisions systematically and integrate succession planning with organizational needs to minimize crises that might occur [21,23].

Formal succession planning for nurse managers must be integrated into the organization’s strategic plan [15]. Succession planning must be part of the strategic plan for every nursing organization [17]. The strategic plan should focus on methods for effectively recruiting potential young leaders [34]. The continuum in succession planning begins with a clear vision and strategic plan, covering recruitment, development, and training of all nurses to achieve new competencies [35].

A study from United States, there is a Nurse Manager Residency Program [36]. This program focuses on organizational leadership competencies as its curriculum framework, so it is designed for residents to gain broad exposure to inpatient care units, specialty departments, and hospital operations. Residents participate in quarterly training sessions hosted by the head nurse. During these sessions, changes to policy and practice are implemented and plans of action are developed, progress with priority strategies is reviewed, including quality indicators of nursing care as well as patient and nurse satisfaction data. At the organizational level, residents attend leadership sessions led by the chief executive officer and the senior leadership team. In addition to financial, HR, and operational agenda items, topics related to ethics and corporate responsibility, servant leadership, healthcare reform, and cultural diversity and inclusion are discussed. nurse manager residency is one way to implement succession planning. In this program, there is evidence of organizational commitment to professional leadership development. This program makes it easier for nurses who are still in transition to becoming nurse managers in terms of their competence and confidence [36]. Residents’ competency and confidence can increase because residents are used to managing nursing management in quarterly training at The Nurse Manager Residency Program.

In another study in the United States, there is a program called Nurse Management Internship [37]. This program is aimed at nurses who are currently completing their BSN and have been working for at least one year as an RN. In the first month, interns complete a total of 20 h of activity including 16 h of management time (in active contact with a manager or director), 2 h of educational time, and 2 h of personal reflection time. Interns also attend events where they gain access to all meetings and are invited to fully participate in discussions and activities with veteran nurse managers aimed at providing important basic information about the nurse manager role and the internship program. In the second to fifth months, interns meet for 2 h of education and discussion of management topics selected from a theoretical framework. Interns also select, read, and appraise one research article about nursing management each month to develop research evaluation skills. It includes time to share experiences, including judgments of leadership style, ways of solving problems, and role insights. Interns are provided with a template of expected internship experience to help them organize and document a diverse internship experience. They were also asked to identify personal and professional goals at the end of the first day. These personal goals and internship goals are used to guide individual activities and to provide a structured evaluation program. During the internship, each intern spends one full day with two different novice nurse managers, two professional nurse managers, and two expert nurse managers. Each participant also spends one day with four different directors, depending on their stated goals. Accordingly, the six month internship is primarily training and actively exploring the role of the nurse manager from a variety of organizational and operational perspectives while observing and working with nurse managers at various stages in their management career. A nurse management internship program provides an appropriate depth and breadth of experience that helps interns understand the role and responsibilities of a nurse manager and is also a source of pride, satisfaction, and even joy for the role of nurse managers [37].

The millennial generation must be prepared to fill the void in the position of a leader or nurse manager. However, the role of nurse manager is very complex, stressful, and not what millennial nurses want. Therefore, it is necessary to have support for the implementation of succession planning so that millennials have the desire to have a leadership role in nursing. The millennial generation needs support from superiors who understand succession planning so they can guide and assist them so they can choose a career in nursing leadership [25].

### 4.4. Limitations of Previous Studies

There are some limitations of previous studies. Researchers only focus on successful candidates’ feedback [11], so that others’ knowledge about succession planning cannot be evaluated. That makes a lack of systematic evaluation of succession planning [15]. They should ask for feedback from all candidates to evaluate the entire study. It is intended that the succession planning program in the future avoids things that interfere with the implementation of the succession planning that has been performed. Furthermore, some studies conducted in some district hospitals in one region of a country or only included a small sample from 1 hospital [12,16], so the findings might not be representative.

## 5. Conclusions

The reviews focus on the factors that influence the effectiveness of succession planning, the benefits of succession planning, and the application of succession planning in clinical practice. We already know that succession planning has many benefits for individual nurses, as well as nursing organizations, and future nursing leadership. Succession planning will indirectly prepare competent and more confident leaders, so that nursing institutions are proactive toward sudden leadership vacancies. This article can be a suggestion for nurse managers to perform succession planning effectively by considering everything that happened in clinical practice. Nurse managers have to improve several things; training and mentoring related to leadership, support from human resources, adequate funding, promotion, communication, cooperation, coordination, and transparency between academics. Nurse managers also have to pay more attention to things that happen in clinical practice; the process of recruitment and planning for nurse managers that occurs in the field is not optimal, so the nurse managers can learn from their mistakes so it does not happen in the future. Nurse manager residencies and nurse management internships can be an option for the application of succession planning in clinical practice. Nurse executives and other nurse leaders must be actively involved in the formal succession planning process for the sake of greater continuity of strategic leadership, operational effectiveness, and quality of care.

## 6. Future Direction

This study suggests other researchers make a study that focuses on how to apply succession planning in clinical practice in a more comprehensive way.

## Data Availability

Not applicable.

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
