# Peer review of "Succession Planning Leadership Model for Nurse Managers in Hospitals: A Narrative Review"

_healthcare, 2023, doi:10.3390/healthcare11040454_

Round 1

Reviewer 1 Report

Dear Authors,

My comments are in the attached PDF, with highlights &/ red crossed lines and comments on the left hand side, or in text boxes.

This review only stated the obvious, which in fact is not unique to nursing - any succession planning is invariably good for standards, morale and organisational success. 

I have made many suggestions for you to improve on the English in the Introduction. The style of writing in the Introduction is vastly different than that of the Discussion section up to sub-section 4.3 and then the previous style re-emerges. You need to standardise the style of writing. Furthermore, the discussion section reads like a regurgitation of the narrative review. It needs to be far less repetitive.

Please update all the use of "head of room". In English we call them Ward Managers or Nurse Managers.

The conclusion is merely a summary of the discussion and doesn't inform the reader on how you might use this information going forward in your clinical setting. In fact, some people might argue that although you've summarised the publications you reviewed and pointed out the obvious about succession planning, you have not really provided a succession planning model, as you set out to do. A model would be like a guideline, not just a collection of observations from the reviewed publications.

Reviewer 2 Report

This study identified the nurse succession planning model and its application in the clinical setting by using a narrative review approach, which will help to provide guidance and assistance for the younger generation who will become leaders in the future. Generally speaking, this article is of practical value. There are some problems, which must be solved before it is considered for publication.

1. The literature review is not extensive but provide the theoretical lens for the narrative study. In short, I would suggest improving it with a critical point of view or an explanation of some limitations of previous studies.

2. Relevant research background needs to be supplemented in INTRODUCTION. The author would better provide the proportion of Indonesian nurses and leaders, and compare it with the world level.

3. Why does the author choose these literature for display? What is the basis for selecting these literature ? Is the literature selection sufficient?

4. The DISCUSSION section is too simple, and there is no new structured knowledge discovery. CONCLUSIONS needs more in it, as it's more of an afterthought. The authors are suggested to highlight important findings and include afterthought of this work. The significance of this paper is not expound sufficiently. The author need to highlight this paper's innovative contributions.

5. Your manuscript needs careful editing and particular attention to English grammar, spelling, and sentence structure.

6. Please check format for the references
